# High-temperature superconductivity on the verge of a structural instability in lanthanum superhydride

Dan Sun [1,8 ✉], Vasily S. Minkov [2,8], Shirin Mozaffari[3], Ying Sun [4], Yanming Ma [4,5,6], Stella Chariton[7], Vitali B. Prakapenka [7], Mikhail I. Eremets [2], Luis Balicas [3] & Fedor F. Balakirev [1]

The possibility of high, room-temperature superconductivity was predicted for metallic hydrogen in the 1960s. However, metallization and superconductivity of hydrogen are yet to be unambiguously demonstrated and may require pressures as high as 5 million atmospheres. Rare earth based "superhydrides", such as $LaH_{10}$, can be considered as a close approximation of metallic hydrogen even though they form at moderately lower pressures. In superhydrides the predominance of H-H metallic bonds and high superconducting transition temperatures bear the hallmarks of metallic hydrogen. Still, experimental studies revealing the key factors controlling their superconductivity are scarce. Here, we report the pressure and magnetic field dependence of the superconducting order observed in $LaH_{10}$. We determine that the high-symmetry high-temperature superconducting *Fm-3m* phase of $LaH_{10}$ can be stabilized at substantially lower pressures than previously thought. We find a remarkable correlation between superconductivity and a structural instability indicating that lattice vibrations, responsible for the monoclinic structural distortions in $LaH_{10}$, strongly affect the superconducting coupling.

[1] National High Magnetic Field Laboratory, Los Alamos National Laboratory, Los Alamos, NM, USA. [2] Max-Planck Institute for Chemistry, Mainz, Germany. [3] National High Magnetic Field Laboratory, Florida State University, Tallahassee, FL, USA. [4] International Center for Computational Method and Software, College of Physics, Jilin University, 130012 Changchun, China. [5] State Key Laboratory of Superhard Materials, College of Physics, Jilin University, 130012 Changchun, China. [6] International Center of Future Science, Jilin University, 130012 Changchun, China. [7] Center for Advanced Radiation Sources, University of Chicago, Chicago, IL, USA. [8] These authors contributed equally: Dan Sun, Vasily S. Minkov. ✉email: ustsundan@gmail.com

For phonon-mediated superconductors, a high transition temperature necessitates light atomic masses. The lightest atom available to compose a crystal lattice is hydrogen, which forms covalently bonded molecular dimmers under ambient conditions. Transforming pure molecular hydrogen, with the aid of pressure, into a metal with an atomic lattice and into a superconductor has been a long-standing challenge and the subject of contention for the high-pressure community. Yet, chemical pre-compression with certain elements reduces the pressure required for metallization; thus, stable hydrogen-rich phases can be synthesized by the current high-pressure technology. With the discovery of a superconducting transition at the critical temperature $T_c = 203$ K in $H_3S$ at 150 GPa[1], the search for hydrogen-rich high-temperature superconductors (HTS) has intensified, with the recent report of room-temperature superconductivity in C-S-H system with a maximum $T_c$ of 288 K[2]. A new family of rare-earth hydrides, such as $LaH_{10}$[3,4] and $YH_9$[5], opened a path to a significant increase in $T_c$, which is predicted to reach 305–326 K in $YH_{10}$[6].

While in $H_3S$ the crystal lattice is formed by H-S covalent bonds, $LaH_{10}$ forms a clathrate-like structure, where each La atom is locked at the center of a $H_{32}$ hydrogen cage. The interatomic distances between hydrogen atoms in $LaH_{10}$ are close to the H–H distance predicted for atomic metallic hydrogen near $p = 500$ GPa[6]. Due to the short H–H distances and the high hydrogen content, $LaH_{10}$ can be considered as "doped" metallic hydrogen. A pronounced isotope effect on $T_c$ when hydrogen is substituted by its heavier isotope deuterium, confirmed that the superconductivity in HTS hydrides is induced by electron–phonon interactions[7]. However, there is a dearth of experimental studies on HTS hydrides due to the very limited number of measurement techniques available at such extreme pressures. Here we explore the superconductivity and the structure of the lanthanum hydride family over a wide range of pressures, temperatures, and magnetic fields. We find that superconductivity in $LaH_{10}$ is strongly affected by a crystal lattice instability toward symmetry-lowering distortions. A similar dramatic change in the $T_c(p)$ dependence for another HTS hydride $H_3S$ was also linked to a structural phase transition[8–11]. The present study firmly establishes the connection between HTS and soft phonon modes that are responsible for the structural instability in hydrides.

## Results and discussion

**The relation between crystal structure and superconductivity.** Metallic lanthanum readily reacts with hydrogen at high pressures and temperatures yielding the clathrate-like superhydride $LaH_{10}$. We found that the superconducting $Fm$-$3m$ phase of $LaH_{10}$ can be synthesized at pressures much lower than $\sim$150–170 GPa[3,4,12,13] as reported earlier. Specifically, the powder X-ray diffraction data show that the sample prepared in the present study under 138 GPa is comprised of the dominant $Fm$-$3m$ phase of $LaH_{10}$. The minor impurity phases are attributed to two hexagonal close-packed (hcp) phases with the $P6_3/mmc$ space group but with a different $c/a$ ratio ($\sim$1.63 for hcp-I and $\sim$1.48 for hcp-II) and a composition close to $LaH_{10}$ (Fig. 1). Both impurity phases were also found in various samples prepared via the direct chemical reaction between hydrogen and lanthanum or lanthanum trihydride in the previous work[3] and did not distinctly affect the $T_c$ of the superconducting $Fm$-$3m$ phase, which has the highest $T_c$ in the lanthanum–hydrogen system[3].

The persistence of the high-pressure, high-symmetry phase of $LaH_{10}$ at pressures as low as 138 GPa corroborates recent theoretical calculations that take quantum effects into account[14]. In contrast to the classical ab initio calculations[6,14,15], which predict structural distortions in the $Fm$-$3m$ $LaH_{10}$ below

$\sim$230 GPa, the inclusion of the zero-point energy stemming from quantum atomic fluctuations lowers the enthalpy of the high-symmetry phase and stabilizes it at pressures as low as 129 GPa[14].

The $LaH_{10}$ sample under 138 GPa still exhibits a narrow superconducting transition toward zero resistance with a high $T_c$ of 243 K, slightly lower than the maximum $T_c$ of $\sim$250 K reported for $LaH_{10}$ at $\sim$150–170 GPa[3,4,13], in accordance with a "dome-shape" pressure dependence of $T_c$ for the $Fm$-$3m$ phase of $LaH_{10}$[3]. No intrinsic hysteresis between cooling and warming $R(T)$ curves was observed (Supplementary Fig. 1). The resistivity $\rho$ of $LaH_{10}$ is estimated to be $(0.3 \pm 0.1)$ m$\Omega$·cm at $T = 300$ K and is higher than the value reported for $H_3S$[16]. The large error bar is mainly due to the uncertainty on the thickness of the sample.

After the abrupt decompression from 138 to 120 GPa, some reflections from the ancestral cubic phase became split (Fig. 1) and the $T_c$ dropped to 191 K (Fig. 2). The powder X-ray diffraction patterns of the new distorted phase can be reasonably indexed within the $C2/m$ space group (Fig. 1b). The refined cell parameters and the coordinates of the heavier La atoms are in a good agreement with theoretical models for the $C2/m$ $LaH_{10}$ phase[14,17,18]. According to the theoretical calculations[14], the monoclinic scenario of the structural distortions is energetically more favorable than two alternative orthorhombic and rhombohedral distortions of the $Fm$-$3m$ phase of $LaH_{10}$ on decompression.

We found that these monoclinic structural distortions are reversible, and the high-symmetry phase can be restored if the pressure is increased. The observed $T_c$ increases rapidly within a short pressure range with increasing pressure and reaches 241 K at 136 GPa (Fig. 2a). The broadening of the superconducting transition in Fig. 2a is likely caused by the deterioration of the phase crystallinity during variations of the pressure. The continuous change of the lattice volume during the $Fm$-$3m$–$C2/m$ phase transition is in close agreement with both the experimental[3] and theoretical[19] equations of state for $LaH_{10}$ indicating the retention of the $LaH_{10}$ composition (Supplementary Fig. 2). In addition, predictions suggest that the composition should not change during any structural distortion scenario for the $Fm$-$3m$ phase of $LaH_{10}$ upon decreasing pressure[14,17,18].

The pressure dependence of $T_c$ in Fig. 2 displays two distinct regions—a low-pressure region characterized by a sharp rise in $T_c$, and a high-pressure region with a much more moderate dome-like $T_c(p)$ dependence, with a clear boundary between the two regions at 135 GPa. This distinct shape in $T_c(p)$ in $LaH_{10}$ closely resembles the $T_c$ variation first discovered in the hydride $H_3S$, where a sharp but continuous drop in $T_c$ was attributed to the change of the crystalline structure[8,10,11]. Multiple distorted hydrogen arrangements from a high-symmetry $Fm$-$3m$ phase are predicted for $LaH_{10}$ as well[14]. One of the predictions reports a stable $LaH_{10}$ $Fm$-$3m$ phase at high pressures, with symmetric H positions and a $T_c$ of 259 K at 170 GPa. The drop in pressure is predicted to stabilize a distorted $R$-$3m$ phase of $LaH_{10}$, with $T_c = 206$ K at 150 GPa[18]. A $T_c \sim 229$–245 K was calculated for the $C2/m$ phase, although the calculations were performed for $p = 200$ GPa, which is substantially higher than the values presented here[17].

**The softening of lattice vibrations.** A likely explanation for the drastic change in the dependence of $T_c$ with pressure <135 GPa is a structural phase transition in $LaH_{10}$. The lack of a discontinuous jump in $T_c$ in $LaH_{10}$ and in $H_3S$[8,10] points to a continuous symmetry-lowering lattice distortion or a phase transition of the second order or weakly first order. We calculated phonon dispersion relations for the high-symmetry $Fm$-$3m$ and distorted $C2/m$ phases of $LaH_{10}$ and identified the lattice vibrations that soften upon decompression and can be linked to the

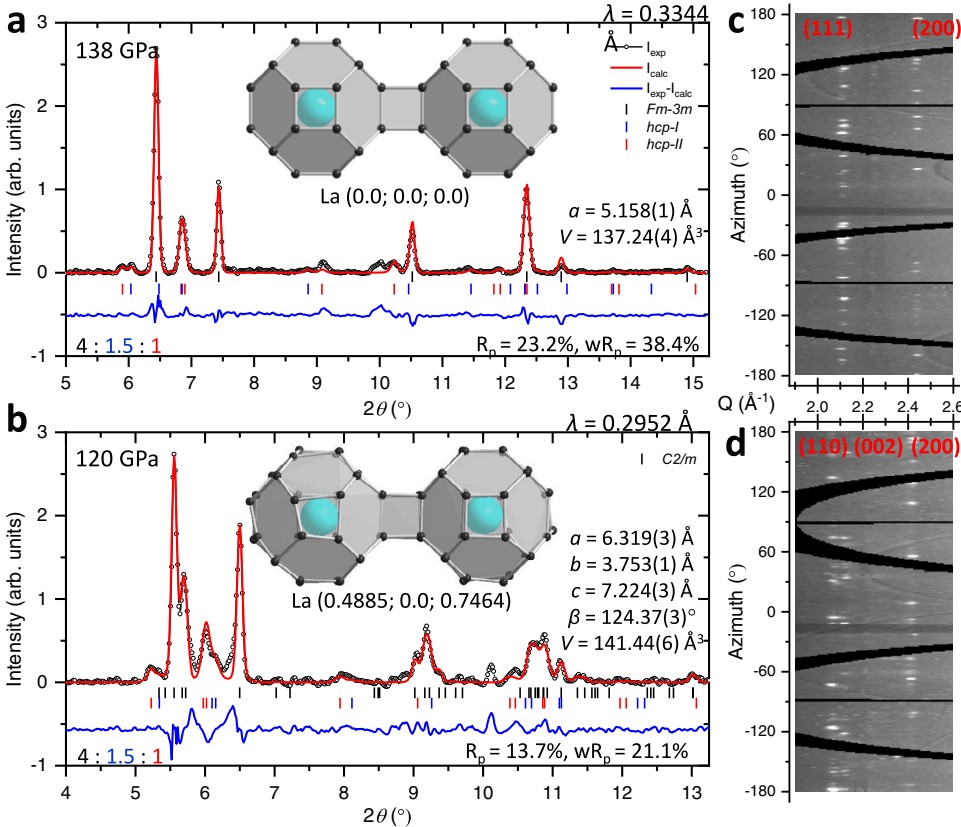

**Fig. 1 Structural data for LaH₁₀ synthesized from La and excess H₂. a, b** Rietveld refinement for *Fm-3m* phase of LaH₁₀ at 138 GPa and *C2/m* phase of LaH₁₀ at 120 GPa, respectively. The peaks originating from the *hcp-I* ($a = 3.668(4)$ Å; $c = 5.914(11)$ Å; $V = 68.9(1)$ Å³ at 138 GPa) and *hcp-II* ($a = 3.750(3)$ Å; $c = 5.561(7)$ Å; $V = 67.7(1)$ Å³ at 138 GPa) impurity phases are indicated through blue and red dashes, respectively. The refined ratio between the main and the impurity phases is provided in the left bottom corner of each figure. The main structural building block, two connected LaH₃₂ polyhedra, are shown in the middle inserts for each phase. Large blue and small black spheres correspond to La and H atoms, respectively. **c, d** The original powder X-ray diffraction patterns at 138 and 120 GPa, respectively. New reflections appear at 120 GPa due to the monoclinic distortions.

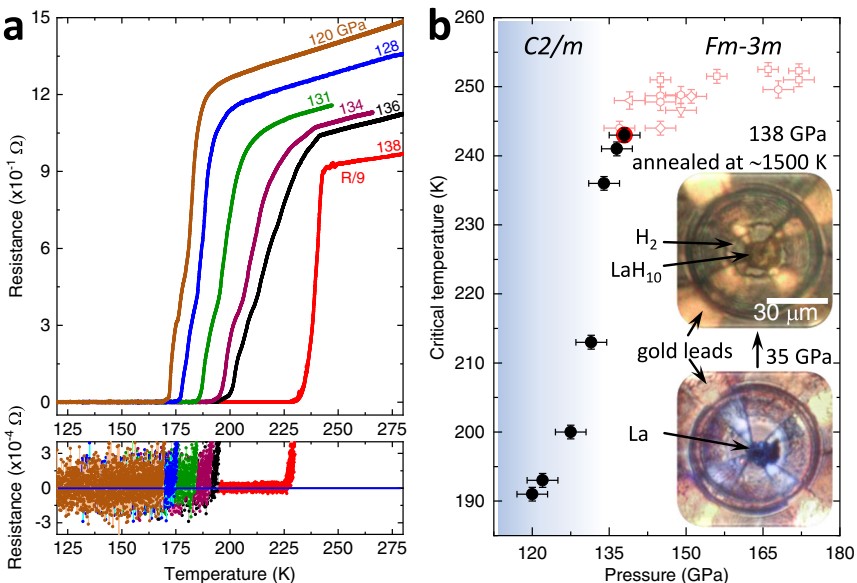

**Fig. 2 The superconducting transitions in LaH₁₀. a** The electrical resistance in LaH₁₀ after the synthesis under 138 GPa (red curve), after the abrupt decompression down to 120 GPa (brown curve), and upon a gradual increase in pressure from 120 to 136 GPa (blue, green, purple, and black curves). The data measured at 138 GPa on the upper panel are divided by 9 for better presentation. **b** Pressure dependence of $T_c$ in LaH₁₀ measured in the present study (black symbols) and from a prior study[3] (open red symbols). Insets: photos of the DAC loaded with a La flake and after the synthesis of LaH₁₀ through laser-assisted heating.

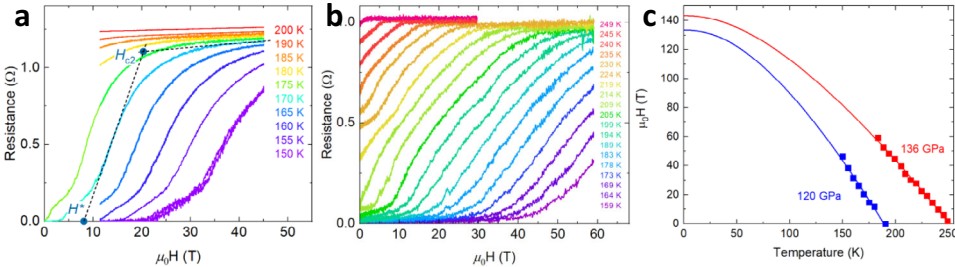

**Fig. 3 Resistance of LaH$_{10}$ as a function magnetic field at different temperatures. a** DC field measurements for the *C2/m* phase of LaH$_{10}$ at 120 GPa. Two dashed lines extrapolate the slope of the high-temperature superconducting transition (left line) toward the asymptotic trace representing the high-field normal state magnetoresistance (right line) at 170 K, respectively. The intersection between two lines provides an estimation of the upper critical field ($H_{c2}$). The intersection of the first line with the horizontal axis indicates the irreversibility field ($H^*$) for the high-temperature superconducting phase. **b** Pulsed field measurements for the *Fm-3m* phase of LaH$_{10}$ at 136 GPa. Both DC and pulsed field traces were recorded under isothermal conditions with no observation of eddy current-generated Joule heating due to the sweeping of the field. **c** Fits of the superconducting upper critical $H_{c2}$ to the Werthamer–Helfand–Hohenberg (WHH) formalism. Red and blue squares denote the loci of $H_{c2}$ of LaH$_{10}$ at 136 and 120 GPa, respectively. Lines of the same color correspond to the WHH fits of the experimental data.

**Table 1 Summary of sample properties and the associated WHH fit parameters: the critical temperature, the upper critical field at *T* = 0 K, coherence length at *T* = 0 K, BCS Fermi velocity, calculated bare-band Fermi velocity, and the slope of $H_{c2}$ at the critical temperature.**

| Sample structure | *p* (GPa) | $T_c$ (K) | $H_{c2}$ (0) (T) | $\xi$(0) (nm) | BCS $v_F$ (×10$^5$ m/s) | Band $v_F$ (×10$^5$ m/s) | $dH_{c2}/dT|_{Tc}$ (T/K) |
|---|---|---|---|---|---|---|---|
| *C2/m* LaH$_{10}$ | 120 | 189 | 133.5 | 1.57 | 2.17 | 3.73 | −1.12 |
| *Fm-3m* LaH$_{10}$ | 136 | 246 | 143.5 | 1.514 | 2.77 | 4.99 | −0.83 |

observed lattice distortion (Supplementary Fig. 4). The calculated phonon dispersions show that the *Fm-3m* phase is dynamically stable at 200 GPa. However, the softening of the low-lying H-H "wagging" vibration modes along the Γ−X direction is found in the phonon spectrum (Supplementary Figs. 4 and 5), which leads to a structural instability toward the monoclinic *C2/m* distortion. The classical harmonic treatment of atomic vibrations for the *Fm-3m* phase of LaH$_{10}$ shows negative phonon frequencies at pressures <180 GPa.

The transformation of the crystallographic structure from a higher- to a lower-symmetry phase is governed by the phonon softening when the frequency of the collective atomic movement approaches zero. Such a drastic change in the phonon modes often has a profound effect on the phonon-mediated superconducting order. A boost in the $T_c$ due to phonon softening in the vicinity of a structural transition has been reported in a number of superconducting families, ranging from Sn nanostructures[20], A15 compounds[21], intercalated graphite[22], ternary silicides[23], and even some elements under pressure[24,25]. The symmetry-lowering distortion in the H sub-lattice in LaH$_{10}$ is driven by the softening of the low-lying H-H vibration modes below 500 cm$^{-1}$ (Supplementary Figs. 4 and 5), leading to a stronger electron–phonon interaction in the *Fm-3m* phase, which is characterized by a coupling constant $\lambda = 2\int_0^\infty \alpha^2 F(\omega)\omega^{-1}d\omega$, where $\omega$ is the phonon frequency, $F(\omega)$ is the phonon density of states, and $\alpha^2$ is an average square electron–phonon matrix element. While the light atomic mass of hydrogen is a necessary requirement for phonon-coupled HTS, $T_c$ is also strongly affected by $\lambda$[26–29], with a peak in $T_c$ predicted for large $\lambda \sim 2$–2.5, which should occur in the vicinity of the lattice instability in HTS hydrides[30].

**Lattice distortion effect on superconducting parameters.** The impact of the structural instability on the key parameters of the superconducting phase, including the upper critical field, $H_{c2}$, and the superconducting coherence length, $\xi$, for the *Fm-3m* and

*C2/m* phases of LaH$_{10}$ was confirmed through magnetotransport measurements. The samples were electrically connected in a van der Pauw configuration (Fig. 2c, inset), making the measurements of both resistivity and Hall effect possible. The LaH$_{10}$ sample under 120 GPa was measured up to 45 T in direct current (DC) magnetic fields, and the LaH$_{10}$ sample under 136 GPa was measured in a 65 T pulsed magnet.

The magnetoresistance (MR) of LaH$_{10}$ collected at fixed temperatures is shown in Fig. 3. Under external magnetic fields, the superconducting transitions span tens of teslas, which correlates with the broadening of the superconducting transition at zero field (Fig. 2a). The normal state MR above $H_{c2}$ is nearly field and temperature independent, with a clear kink at the onset of superconductivity at $H_{c2}$. For the consistency with prior studies, the $H_{c2}$ values are determined as the intersection between the straight line extrapolations of the normal state MR and the slope of the superconducting transition by a method similar to the one followed in ref. [16]. The irreversibility field of the high-temperature superconducting phase ($H^*$) is taken by extrapolating the leading edge of the transition to the horizontal axis (Supplementary Fig. 6). The Hall resistance signal measured above $T_c$ is consistent with the electron-like Fermi surface (Supplementary Fig. 7).

Upper critical field measurements in H$_3$S HTS hydride have independently verified a large $\lambda \sim 2$[16]. We find a substantially larger $H_{c2}$ for LaH$_{10}$ and determined that magnetic fields of the order of 100 T are required to distinguish between a strongly coupled scenario with a large $\lambda$ and the more commonly employed Werthamer–Helfand–Hohenberg (WHH) model derived in the weakly coupling limit, $\lambda \ll 1$[31]. To extract the key superconducting properties of LaH$_{10}$ and explore the effects of the structural transition on its superconductivity, we fit the temperature dependence of $H_{c2}$ to the WHH (Fig. 3c and Table 1) formalism. The WHH model fits our data well for fields up to 60 T, which is our upper measurement limit. The WHH model considers the combined effects of the magnetic field on the orbital motion and on the spin of the electrons: $H_{c2}^{-2} = H_{orb}^{-2} + H_p^{-2}$,

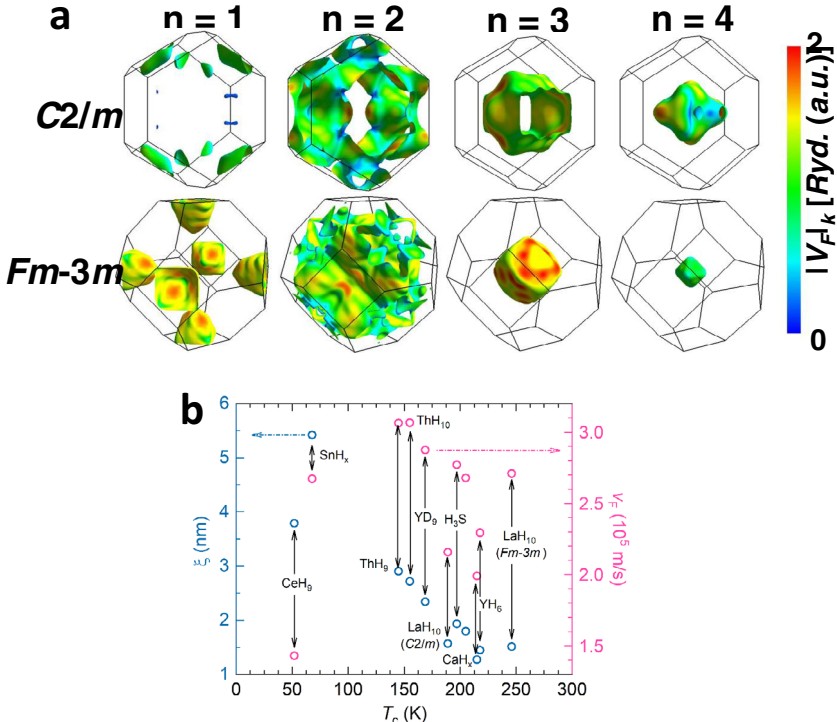

**Fig. 4 Fermi velocities for the different hydrides. a** Calculated Fermi velocities associated with the electronic states on the Fermi surfaces in the first Brillouin zone (frame) where the color scale goes from blue (slowest) to the red (fastest). The perspective comprises angles of rotation with respect to the $x$-, $y$-, $z$-axis of 0°, 0°, and 0° for the $C2/m$-LaH$_{10}$ and 13°, −13°, and 1° for the $Fm$-$3m$-LaH$_{10}$, respectively. **b** Extracted coherence lengths $\xi$ (cyan circles, left axis) and BCS Fermi velocities $v_F$ (magenta circles, right axis) for the different hydrides[5,13,19,41–43]. The labels associated with the $C2/m$-LaH$_{10}$ and $Fm$-$3m$-LaH$_{10}$ phases correspond to results from the present study.

where $H_{orb}$ and $H_p$ are the orbital-limited and spin-limited (Pauli) critical fields, respectively. We obtain $H_p(0)$ values of 352 T at 120 GPa and 457 T at 136 GPa. $H_p(0)$ values are larger by a factor of ~3 when compared to the $H_{c2}(0)$ values listed in Table 1, indicating predominantly orbital-limited upper critical fields in HTS LaH$_{10}$, which is similar to H$_3$S[16].

The WHH fit provides a reasonable estimate of the superconducting coherence length $\xi = \sqrt{\phi_0/2\pi H_{c2}}$, where $\phi_0$ is the magnetic flux quantum. There is a significant drop in $T_c$ in the distorted phase of LaH$_{10}$ at 120 GPa when compared to the LaH$_{10}$ sample at 136 GPa. Surprisingly, $H_{c2}(0)$ only drops by a small amount and thus $\xi$ (0) remains nearly unchanged. $\xi$ is linked to both $T_c$ and the Femi velocity $v_F$: $\xi = 0.18\,\hbar v_F/k_B T_c$ within the BCS theory[7], but the $\xi \sim v_F/T_c$ rule should remain valid for other models, thus signaling a lower value for $v_F$ in the $C2/m$ phase at 120 GPa when compared to that in the $Fm$-$3m$ phase at 136 GPa. The onset of the lattice distortion is expected to be strongly affected by the electron dispersion, e.g. via the flattening of the bands at the boundaries of the new Brillouin zone, which may lead to a drop in $v_F$ so that $\xi$ and $H_{c2}$ remain high despite the drop in $T_c$ in the $C2/m$ phase. We calculated the $v_F$ values along the Fermi surfaces in the first Brillouin zone for both $C2/m$ and $Fm$-$3m$ phases (Fig. 4a). The average calculated $v_F$ values are listed in Table 1 alongside with the BCS values obtained from $H_{c2}(0)$. The calculated $v_F$ values are larger than BCS ones mainly because the calculations do not account for the renormalization of the bare-band $v_F$ due to electron–phonon coupling. Nevertheless, the model provides a more reliable estimate of the relative change in $v_F$. The calculations confirm a ~30% drop in $v_F$ in the $C2/m$ phase as observed in the high-field experiments. A comparative review of the $v_F$ values for other hydrogen-rich

HTS families, which can be extracted from the published $H_{c2}$ data, as well as the present study, reveal a surprisingly narrow distribution close to ~$2.5 \times 10^5$ m/s (Fig. 4b). A similar universal Fermi velocity was first noticed in the HTS cuprates, with a surprisingly similar average value of ~$2.7 \times 10^5$ m/s[32]. This similarity points to a renormalization of the charge carrier band dispersion both in the cuprates and in the hydrides via a strong coupling to low-lying excitations near the Fermi level, the same coupling that is commonly considered to be responsible for the high-temperature superconductivity.

In conclusion, we have measured the properties of the superconducting LaH$_{10}$ compound as a function of pressure, temperature, and high magnetic fields. We find evidence for a pressure-induced $Fm$-$3m$−$C2/m$ structural transition in LaH$_{10}$ at $p_c = 135$ GPa, resulting in a steep but continuous decrease in $T_c(p)$ below $p_c$. A likely mechanism for the structural instability is phonon softening associated with a gradual distortion of the lattice, as proposed for another HTS hydride H$_3$S. We established key superconducting quantities of superhydrides under high magnetic fields, including upper critical fields and coherence lengths. We found that the drop in the Femi velocity in LaH$_{10}$ is consistent with the distortion-induced changes in the Brillouin zone. The proximity of a peak in $T_c$ to a symmetry-lowering structural transition, which is now experimentally established for at least two HTS hydride families, indicates that the tuning of the soft phonon modes should be viewed as one of the main pathways toward maximizing $T_c$ in the hydrogen-rich superconductors.

## Methods
**Diamond anvil cell**. The superconducting sample of LaH$_{10}$ were synthesized in situ in a miniature diamond anvil cell (DAC) with a maximum diameter of 8.8 mm and a body length of ~30 mm. The DAC was designed by reworking and modifying the

prototype piston/locking nut design and briefly discussed in ref. [1]. To minimize the heating effect under high magnetic fields and provide a high mechanical strength, the body of the high-pressure cell was made of a high-purity Cu–Ti alloy with 3 wt % Ti and Cu–Be alloy with 1.8–2.0 wt % Be. These DACs allow one to reach high pressures up to 200 GPa, perfectly fit the narrow bore of high field DC and pulsed magnets, and still have wide opening-angle appropriate to the routine X-ray diffraction and spectroscopic measurements.

**Sample preparation.** For the sample synthesis, a small piece of metallic lanthanum (Alfa Aesar, 99.9%) with a lateral dimension of about 10 μm and a thickness of ~1–2 μm was placed in the center of the beveled diamond anvil with a culet size of 35 μm onto the tips of four sputtered leads. The 120-nm-thick tantalum leads covered by a 50-nm-thick gold layer were sputtered onto the diamond surface in a van der Pauw configuration with a distance of ~4 μm between tips (see Supplementary Fig. 7 inserts). The electrical leads were thoroughly isolated from the metal rhenium gasket by a protecting layer made from magnesium oxide, calcium fluoride, and epoxy glue mixture. Excess hydrogen (Westfalen, 99.999%) was introduced in the DAC at a gas pressure of about 150 MPa and served as both a reactant and a pressure-transmitting medium. After the cell was thoroughly clamped, the sample was pressurized to a pressure of 138 GPa and then heated up to ~1500–2000 K by a microsecond pulse YAG laser to initiate the chemical reaction between reactants. Hydrogen was always in excess, and its presence in the DAC throughout the experiment was monitored visually and by Raman spectroscopy. The pressure was estimated from the Raman shift of the diamond line edge[33] and the vibron of $H_2$[34]. Although both scales indicated the same pressure values within an error of ±5 GPa, we used the hydrogen scale throughout the manuscript.

After electrical transport and X-ray diffraction measurements for the sample of the $Fm$-$3m$ $LaH_{10}$ at 138 GPa, the pressure in the DAC abruptly dropped to 120 GPa during transportation. Then the pressure was increased stepwise up to 136 GPa. Zero field electrical transport properties were measured at each pressure step. X-ray diffraction data were collected at 138 and 120 GPa and magnetotransport measurements under external magnetic fields were done at 120 and 136 GPa.

**Structure characterization.** X-ray diffraction data were collected at the beamline 13-IDD at GSECARS, Advanced Photon Source using $\lambda_1 = 0.2952$ Å and $\lambda_2 = 0.3344$ Å, beam spot size of ~3 × 3 μm, and Pilatus 1 M CdTe detector. Typical exposure time varied between 10 and 300 s. Processing and integration of the powder X-ray diffraction patterns were carried out using the Dioptas software[35]. Indexing and Rietveld refinement were performed in GSAS and EXPGUI packages[36,37]. The coordinates of the heavier lanthanum atoms were refined from the experimental data, whereas H atoms were placed in the theoretically calculated positions. With the lattice parameters and La atom positions fixed, structural relaxation for H atom positions was carried out using the Quantum-ESPRESSO package[38]. Structural data for the refined $Fm$-$3m$ and $C2/m$ phases of $LaH_{10}$ can be obtained as Crystallographic Information Files from the Cambridge Crystallographic Data Centre via www.ccdc.cam.ac.uk/data_request/cif, on quoting the Deposition Number: 2033292–2033293.

The present structural data contradict the antecedent experimental work[12], in which the $Fm$-$3m$ phase of $LaH_{10}$ was studied upon decompression from 169 to 27 GPa. Geballe et al. found a phase transition that breaks the face-centered cubic symmetry of the crystal lattice at pressures of 152–121 GPa and a subsequent decomposition of $LaH_{10}$ compound at lower pressures. The authors proposed an $R$-$3m$ structural model for the distorted phase of $LaH_{10}$, though the observed peaks in powder X-ray diffraction patterns did not match the calculated positions for this model. The same inconsistency between the observed and calculated peaks can be seen if one fits the present X-ray diffraction data at 120 GPa within the $R$-$3m$ model (Supplementary Fig. 2), which indicates that the suggested $R$-$3m$ model is erroneous.

We also found that the instability of the $Fm$-$3m$ $LaH_{10}$ phase occurs at much lower pressures than was claimed by Geballe et al.[12] This discrepancy likely stems from the different pressure scales, which were used for estimation of the pressure values in samples. The hydrogen scale used in the present study is more accurate in comparison with the diamond scale, which is based on the stresses in diamonds and strongly depends on the arrangement of the experiment. Hydrogen is very soft even in the solid state and uniformly transmits pressure over the sample. Therefore, the peak of the hydrogen vibron is sharp and the corresponding pressure values can be determined with accuracy greater than ±3 GPa. The uncertainty of the estimation of pressure values in DAC using the diamond scale is shown in Supplementary Fig. 3b. Based on the spectroscopic study of 21 various samples of $D_2$, the dispersion of the pressure values estimated using the diamond scale for the same positions of the high-wavenumber $D_2$ vibron can be as high as 25 GPa in the pressure range of 100–200 GPa. The similar difference of ~18 GPa is observed between the present data and data from ref. [12] on the pressure dependence of the volume per La atom in $LaH_{10}$. Our data are in very close agreement with the equation of state of $LaH_{10}$ calculated using the Quantum Espresso pseudopotentials[19]. Thus, the overestimated pressure value of 152 GPa, which was claimed as the beginning of the structural instability of the $Fm$-$3m$ phase of $LaH_{10}$ in ref. [12] should be reduced to ~134 GPa, which is in perfect agreement with $p_c = 135$ GPa found in the present study.

**Zero field electrical transport measurements.** Zero field electrical resistance was measured through a four-probe technique in van der Pauw geometry with currents ranging from $10^{-4}$ A at $p = 138$ GPa to $10^{-3}$ A at $p = 120$–136 GPa samples. No apparent effect of the current value on the measured $T_c$ was observed. The electrical measurements are presented in a warming part of a thermal cycle as it yields a more accurate temperature reading: the sample is warmed up slowly (0.2 K/min) under nearly isothermal environmental conditions (no coolant flow). The temperature was measured by a Si diode thermometer attached to the DAC with an accuracy of ~1 K. $T_c$ was determined at the offset of superconductivity—at the point of apparent deviation in the temperature dependence of the resistance from the normal metallic behavior.

**Magnetotransport measurements.** MR and Hall effect measurements under high magnetic fields were conducted in the 45 T hybrid magnet and in the 65 T pulsed magnet at the National High Magnetic Field Laboratory. A copper thermal shield was placed around the DAC during DC field measurements. The thermal shield was heated uniformly to reduce the thermal gradients, and a secondary Cernox thermometer was attached to the DAC gasket for accurate measurements of the sample temperature. There is no observable heating from the ramping of the magnetic field at rates up to 3 T/min. The Hall effect was measured for the sample at 120 GPa above $T_c$ in the hybrid DC magnet from 11.5 to 45 T. Reverse-field reciprocity method was employed to determine Hall resistance $R_{xy}$[39] because the field direction of the hybrid magnet cannot be reversed during the day shift. A high-frequency (290 kHz) lock-in amplifier technique was employed to measure sample MR in 65 T pulsed magnet. AC current 500 μA was applied to the sample, and the voltage drop across the sample was amplified by an instrumentation amplifier and detected by a lock-in. No sample heating was observed during ~50 ms long magnet pulse based on comparisons of up sweep and down sweep resistance traces at different field sweep rates.

**Computational details.** Structural relaxation, electronic structures, and phonon calculations were carried out within the framework of density functional theory (DFT) as implemented in the Quantum-ESPRESSO package[38]. Structure relaxations were performed using DFT using the Perdew–Burke–Ernzerhof generalized gradient approximation[40]. Phonon dispersion calculations were performed with the density functional perturbation theory. Ultrasoft pseudopotentials for La and H were used with a kinetic energy cutoff of 80 Ry. To reliably calculate the phonon dispersion, we have employed dense $k$-meshes and $q$-meshes for all the phonon calculations: $8 \times 8 \times 4$ $k$-meshes and $4 \times 4 \times 2$ $q$-meshes for the $C2/m$-$LaH_{10}$ structure and $12 \times 12 \times 12$ $k$-meshes and $6 \times 6 \times 6$ $q$-meshes for the $Fm$-$3m$-$LaH_{10}$ structure. The visualization of the atomic vibrations was done by using a visualization tool (http://henriquemiranda.github.io/phononwebsite/phonon.html). For visualization in three-dimensional of the Fermi velocities associated with the electronic states on the Fermi surfaces in the first Brillouin zone, we have used FermiSurfer open software drawing code (http://fermisurfer.osdn.jp/).

**WHH model.** Numerical fit to the WHH model for the temperature dependence of $H_{c2}(T)$ defined by orbital and spin-paramagnetic effects in the dirty limit is given by WHH[31]:

$$\ln\left(\frac{1}{t}\right) = \sum_{\nu=-\infty}^{\infty}\left\{\frac{1}{|2\nu+1|} - \left[|2\nu+1| + \frac{\bar{h}}{t} + \frac{(\alpha\bar{h}/t)^2}{|2\nu+1| + (\bar{h}+\lambda_{SO})/t}\right]^{-1}\right\} \quad (1)$$

where $\bar{h} = (4/\pi^2)[H_{c2}(T)/T_c(-dH_{c2}/dT)_{T_c}]$, $\alpha$ is the Maki parameter, and $\lambda_{SO}$ is the spin–orbit constant. The Maki parameter for each sample is estimated from the slope of $H_{c2}(T)$ at $T = T_c$: $\alpha = \sqrt{2}\,H_{c\,orb}/H_{c\,p} \sim -0.52758dH_{c2}/dT|_{T_c}$ [26].

## Data availability
The data that support the findings of this study are available in Open Science Framework with the identifier: https://doi.org/10.17605/OSF.IO/RUJWA. Source data are provided with this paper.

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

## Acknowledgements

The synchrotron X-ray diffraction data were collected at GeoSoilEnviro CARS (The University of Chicago, Sector 13), Advanced Photon Source (APS), Argonne National Laboratory (USA). GeoSoilEnviro CARS is supported by the National Science Foundation-Earth Sciences (EAR-1634415) and Department of Energy-GeoSciences (DE-FG02-94ER14466). This research used resources of the Advanced Photon Source, a U.S. Department of Energy (DOE) Office of Science User Facility operated for the DOE Office of Science by Argonne National Laboratory under Contract No. DE-AC02-06CH11357. The work performed at the National High Magnetic Field Laboratory is supported by the National Science Foundation Cooperative Agreement No. DMR-1644779 and the State of Florida. L.B. is supported by the Department of Energy, Basic Energy Sciences through award DE-SC0002613. M.I.E. acknowledges great support from the Max Planck Society.

## Author contributions

D.S., V.S.M. and F.F.B. designed the research and wrote the paper; V.S.M. prepared the samples, collected synchrotron X-ray diffraction data, performed electrical transport measurements without external magnetic field, and processed the structural data; D.S., F.F.B., S.M., L.B. and M.I.E. performed electrical transport measurements under external magnetic fields and processed the data; D.S., F.F.B., S.C. and V.B.P. assisted with the synchrotron X-ray diffraction experiments; Y.S. and Y.M. performed structural relaxation, electronic structures, and phonon calculations. M.I.E. designed the diamond anvil cell. All authors contributed to writing the paper.

## Competing interests

The authors declare no competing interests.
