## [Peer Review File · Nature Communications]

REVIEWER COMMENTS

Reviewer #1 (Remarks to the Author):

This manuscript reports an experimental study of the structural and superconducting properties of lanthanum super hydrides. The major contribution is the extension of the magnetic field effect on superconductivity up to 45 and 60 T for the different superconducting phases. The authors carefully check the structural evolution path with pressure. A new superconducting phase diagram as functions of temperature, pressure, and magnetic fields is established. Basically, it is a nice work and might be recommended for publication in Nature Communications if the following points can be fixed,

1) The significance of this work is absent. The determination of the upper critical field is important for drawing some information such as the superconducting coherence length based the theoretical model. However, the 45 T for the low-pressure phase and 60 T for the high-pressure one are still not high enough to approach the limit at the absolute zero temperature. The low-field results have been published in Ref. [3]. The parallel work for H₃S has also been published by the same team in Ref. [13]. The authors do not add new understanding beyond their early publications,

2) Many paragraphs of this manuscript are used to present the structural data and the implication of these results. The diffraction patterns shown in Fig. 1 are not good enough to draw the accurate atom positions, occupancies, and coordinations. For the accurate structural determination, the growth of the single crystals is needed. In fact, the high-pressure structures were reported previously in Refs. [3,12]. The authors are invited to demonstrate how good their data are compared with the early studies and why this needs to be published in NC.

3) The transition pressure from the different phases is lower than the early report (Ref. 12). The authors believe that this is probably due to the good pressure environment created by the extra hydrogen. No evidence is provided to support this idea in the manuscript.

4) The important claim of this work is the T_c enhancement due to the appearance of the soft mode near the structural instability. Although it might be true, it does not get support from their current results. For the justification, they can either perform the calculations of the phonon dispersions at the interested pressure based on their EOS or collect the phonon spectra at high pressures. In either way, the solid conclusion can be drawn.

5) The early or relevant works are not well acknowledged. For examples, the magnetic susceptibility for LaH₁₀ and the summary for the soft mode picture for super hydrides have been reported in Matter Radiat. Extremes 5, 028201 (2020); 5, 068102 (2020).

6) The manuscript is not prepared in good shape. It does not follow well. Many typos remain.

Overall, this manuscript is a good try to uncovering the hidden physics of superconductivity in super hydrides. I would invite the author to improve the quality of this work.

Reviewer #2 (Remarks to the Author):

Sun et al. reported a high-temperature superconducting phase with a $T_c \sim 243$ K at 138 GPa in LaH10. The authors determined upper critical fields and coherence lengths for this compound. They also provided evidence for a pressure-induced structural transition at 135 GPa. It was suggested that a possible mechanism for the structural instability is phonon softening and a gradual distortion of the lattice. The findings in this manuscript are important and help to shed light on the understanding of the high T_c discovered in the superhydrides. However, there are a few points the authors need to address before I recommend the publication of this work in Nature Communications.

1. From material preparation point of view, what is the difference between the LaH10 reported in this paper and previous work? Why the high T_c can be stabilized at substantially lower pressure values than previous studies.
2. Did the authors measure RT during decompression from 138 GPa to 120 GPa? If so, how does the T_c change as a function of pressure?
3. Can the authors present at least one set of cooling and warming curve under the same pressure? Is there any thermal hysteresis? If there is, is the hysteresis intrinsic?
4. Can the authors provide one set of data to show the magnetic field effect on T_c ?
5. Why the resistance of LaH10 under 138 GPa is much larger than the others measured between 120 GPa and 136 GPa?
6. What is the material for the dac? Did the authors use the same type of dac for the magnetotransport measurements and X-ray measurements?
7. What is the rhenium gasket dimension? What is the sample space dimension?
8. The authors mentioned, "no apparent effect of the current value on the measured T_c was observed". However, in the reference Phys. Rev. Lett. 122, 027001 (2019), M. Somayazulu et al. showed a clear effect of I on the T_c for LaH10. Could the authors comment on this difference?
9. On the last paragraph of Page 4, the authors discussed phonon softening and coupling constant. Can the authors provide a more direct analysis on this part by using the detailed values related to LaH10?

Reviewer #1 (Remarks to the Author):

This manuscript reports an experimental study of the structural and superconducting properties of lanthanum super hydrides. The major contribution is the extension of the magnetic field effect on superconductivity up to 45 and 60 T for the different superconducting phases. The authors carefully check the structural evolution path with pressure. A new superconducting phase diagram as functions of temperature, pressure, and magnetic fields is established. Basically, it is a nice work and might be recommended for publication in Nature Communications if the following points can be fixed,

1) The significance of this work is absent. The determination of the upper critical field is important for drawing some information such as the superconducting coherence length based the theoretical model. However, the 45 T for the low-pressure phase and 60 T for the high-pressure one are still not high enough to approach the limit at the absolute zero temperature. The low-field results have been published in Ref. [3]. The parallel work for H₃S has also been published by the same team in Ref. [13]. The authors do not add new understanding beyond their early publications,

This study provides the first comprehensive characterization of superconducting properties and crystal structure of LaH₁₀ in a finely tuned range of pressures below 150 GPa, by a combination of synchrotron X-ray diffraction, resistivity, and high field magnetotransport measurements with the support from theoretical calculations. The data themselves are highly sought by the community and the existence of H₃S data cannot diminish the significance of the current study, since H₃S and LaH₁₀ belong to compounds with different chemical bonding. In particular, the structural and electronic properties of H₃S are governed by S-H covalent bonds while properties of LaH₁₀ are largely governed by H-H metallic bonds. In contrast to H₃S, we found that the superconducting *Fm-3m* LaH₁₀ phase is much more robust – it is stable at pressures significantly lower than the earlier reported values of 150-160 GPa, still exhibiting high critical temperature (T_c).

For the first time, we explore the relationship between the crystal structure and the key superconducting parameters such as the T_c , the upper critical field, and the coherence length. We link a significant boost in T_c to structural instability towards a symmetry-lowering phase transition. With further support from the theoretical modeling, we establish that the structural instability of the high-symmetry cubic phase is associated with the softening of the lattice vibrations and enhancement of phonon-mediated superconducting coupling in LaH₁₀, which provides a guidance for maximizing T_c in hydride superconductors.

2) Many paragraphs of this manuscript are used to present the structural data and the implication of these results. The diffraction patterns shown in Fig. 1 are not good enough to draw the accurate atom positions, occupancies, and coordinations. For the accurate structural determination, the growth of the single crystals is needed. In fact, the high-pressure structures were reported previously in Refs. [3,12]. The authors are invited to demonstrate how good their data are compared with the early studies and why this needs to be published in NC.

We have shortened the discussion of the structural data in the main text and moved a relevant part to the Method section. The revised version should be more accessible to the general reader.

We agree that single-crystal XRD is more accurate method for the precise crystal structure refinement; however, the growth of a single crystal is extremely challenging at very high-pressures of about 150 GPa. In our experience with high-pressure and high-temperature synthesis of more than 100 samples of various hydrides including La-H and Y-H, we have never observed a single crystal or several single-crystals forming under such conditions. In addition, the deposited gold leads for electrical transport measurements make the high-temperature recrystallization process even more complicated; they touch the sample and can be easily damaged during laser heating.

On the other hand, there is no evident advantages of single-crystal XRD for hydrogen-rich compounds: hydrogen atoms do not scatter X-rays and thus the coordinations and occupancies cannot be directly refined from the experiment no matter one use single crystal or powder XRD. For such purposes one could use neutron diffraction, but this technique requires large volume of samples, which currently cannot be produced at megabar pressures.

Conversely, powder XRD technique is commonly used for study of materials in DACs at high pressures. The structure of the heavier element (e.g. S, Y, La) sub-lattice is typically determined directly from the experimental data, and then the positions of the hydrogen atoms are inferred by minimizing the total lattice energy using available open-source software packages. This approach has been widely used without significant contention in the literature. For the high-symmetry phases there are not many options for the heavier atom positions in the lattice of a certain space group. In the low-symmetry phases, the situation is more difficult and therefore good experimental data are needed. The distortion of the bcc H_3S lattice on decompression is a good example when the good XRD data are very crucial. In the figure bellow, one can see two different experimental data for bcc H_3S on decompression synthesized from disproportionated H_2S published by Einaga et al. (Nature Physics 2016, the left panel) and from elemental S and D_2 published by Minkov et al. (Angew. Chemie 2020, the right panel). The rhombohedral distortions of the sulfur sublattice in D_3S are pronounced and evident on the XRD powder patterns, only in the latter which has more accurate experimental data,

[Redacted]

The most significant improvement, compared with the previous studies of LaH₁₀ at high pressures, is that we accurately measured the structure of the lower-symmetry phase, which was never well determined before. The available experimental data for LaH₁₀ compound on decompression was published by Geballe et al. (Angew. Chemie 2018). Unfortunately, these data are of poor quality: the diffraction peaks from the sample are broadened (most likely because of poor crystallinity) and highly contaminated by parasitic diffraction peaks from the material of gasket (W and WH). This is likely the reason why Geballe et al. could not precisely determine the character of structural distortions. In addition, the authors did not provide the parameters of refinement that could show the accuracy of the chosen *R-3m* structural model. The lattice parameters were refined by using only first three peaks, and the one of them (the third reflection (102)) does not correspond to the proposed model (see the figure below or Figure 5a in the original paper). This indicates that the *R-3m* lattice is not correct. According to the recent theoretical calculations (Errea et al. 2020) the monoclinic distortions are also energetically more favorable than rhombohedral (*R-3m*) and orthorhombic (*Immm*) distortions.

[Redacted]

3) The transition pressure from the different phases is lower than the early report (Ref. 12). The authors believe that this is probably due to the good pressure environment created by the extra hydrogen. No evidence is provided to support this idea in the manuscript.

We added two new figures in the supplementary materials for better understanding of the problem, which appears during pressure estimation in DAC. We also changed the text in the manuscript.

The estimation of pressure values in DAC can be done by different techniques. Usually one can use Raman spectroscopy or XRD by following the shift of the diamond edge (diamond scale) or hydrogen vibron (hydrogen scale) in Raman spectra or equation of state (EoS) for gasket materials measured by XRD. These scales have different accuracy, especially when the pressure medium is not hydrostatic.

We used hydrogen both as a reactant for hydrogenation chemical reaction and as “soft” hydrostatic medium that provides uniform pressure distribution around sample (see Figure 2 c in the manuscript). According to our data (see figure S3), the diamond scale is not very accurate even for the samples with pure hydrogen, i.e. for the best hydrostatic case. The dispersion of the pressure values estimated for as many as 21 samples of pure D₂ using the diamond scale reaches ~25 GPa for the same position of D₂ vibron in Raman spectra at the pressure range of 100-200 GPa. The uncertainty of the diamond scale must be even higher for “dry” samples with no pressure-transmitting medium.

Geballe et al. [12] estimated pressure values on decompression using diamond scale and lattice parameters of tungsten gasket at the edge of gasket hole. The authors did not provide information about hydrogen vibron after laser heating. One can suppose that hydrogen was completely absorbed by La sample and the gasket material (tungsten) during heating. The photo of the sample with the dramatically shrunk hole after laser heating and the dominant phase of WH observed in the XRD powder patterns, supports this assumption.

[Redacted]

Since the material of gasket readily reacts with hydrogen forming WH, it is also not clear from which particular place of the diamond culet (how far away from the synthesized LaH₁₀ sample) the authors collected XRD of pure W for pressure estimation. It is known that pressure increases towards to the culet edge in “dry” samples.

We plotted the volume per La atom vs pressure for the present data, data from Drozdov et al. (pressure was estimated according to H₂ vibron, Nature 2019) and data from Geballe et al. It is clearly seen that for the same lattice volumes of LaH₁₀ the pressure values estimated by using diamond scale or EoS of W (black circles) in “dry” samples are persistently higher than those estimated by using the hydrogen scale in samples surrounded by H₂ medium (red and

blue circles). Moreover, the present data are in close agreement with the EoS of LaH₁₀ theoretically predicted using the Quantum ESPRESSO pseudopotentials (Semenok, Arxiv 2021). Thus, we conclude that the pressure values are overestimated in work of Geballe et al.

4) The important claim of this work is the T_c enhancement due to the appearance of the soft mode near the structural instability. Although it might be true, it does not get support from their current results. For the justification, they can either perform the calculations of the phonon dispersions at the interested pressure based on their EOS or collect the phonon spectra at high pressures. In either way, the solid conclusion can be drawn.

We modeled LaH₁₀ systems with different symmetries and found a trend of the softening of certain phonons that makes the *Fm-3m* structure unstable toward *C2/m* distortion on decompression. We also calculated the average Fermi velocity in *Fm-3m* and *C2/m* crystal symmetry and found a ~25% decrease in distorted phase that matches the decrease observed in our high magnetic field experiment. The results of the calculations are presented in the manuscript.

5) The early or relevant works are not well acknowledged. For examples, the magnetic susceptibility for LaH₁₀ and the summary for the soft mode picture for super hydrides have been reported in Matter Radiat. Extremes 5, 028201 (2020); 5, 068102 (2020).

We are indeed familiar with the reports in Matter Radiat. Extremes 5, 028201 (2020); 5, 068102 (2020). We agree with the referee comment that this earlier work should be cited in the manuscript for comprehensive review of the prior work on the subject.

6) The manuscript is not prepared in good shape. It does not follow well. Many typos remain.

We rewrote the manuscript, with special attention of the spellings and grammar.

Overall, this manuscript is a good try to uncovering the hidden physics of superconductivity in super hydrides. I would invite the author to improve the quality of this work.

Reviewer #2 (Remarks to the Author):

Sun et al. reported a high-temperature superconducting phase with a $T_c \sim 243$ K at 138 GPa in LaH₁₀. The authors determined upper critical fields and coherence lengths for this compound. They also provided evidence for a pressure-induced structural transition at 135 GPa. It was suggested that a possible mechanism for the structural instability is phonon softening and a gradual distortion of the lattice. The findings in this manuscript are important and help to shed light on the understanding of the high T_c discovered in the superhydrides. However, there are a few points the authors need to address before I recommend the publication of this work in Nature Communications.

1. From material preparation point of view, what is the difference between the LaH₁₀ reported in this paper and previous work? Why the high T_c can be stabilized at substantially lower pressure values than previous studies.

The material preparation follows the same procedure with previous work (Ref. 3). The high T_c phase with lower pressure limits is mainly achieved with the new experiment design: On one hand, previous work focuses on the highest achievable T_c near the “optimally” predicted pressure (Ref. 3). The electrical transport properties and crystal structure of LaH₁₀ at lower pressures were not carefully measured. Thus the boundary and the role of structural instability was never known exactly. On the other hand, we found a ~ 20 GPa difference in the pressure values between previous study (Ref. 12) and the present data, because of the different pressure scales. With 20 GPa adjustment, the phases in current study find good agreement with the stability range of Fm-3m LaH₁₀ phase theoretically calculated by taking the quantum effects into account (Errea et al. Nature 2020, Ref.14).

2. Did the authors measure RT during decompression from 138 GPa to 120 GPa? If so, how does the T_c change as a function of pressure?

The sudden decompression from 138 GPa to 120 GPa was an unexpected change during sample transport to the High Field facility. The temperature dependence of resistance on decompression was measured at end values. Subsequent pressurization from 120 GPa to 136 GPa shows that similar pressure leads to similar T_c 's. We believe that the pressure dependence of the T_c will be independent of the pressurization history.

3. Can the authors present at least one set of cooling and warming curve under the same pressure? Is there any thermal hysteresis? If there is, is the hysteresis intrinsic?

We provide the cooling and warming curves for the 120 GPa sample and present additional experimental considerations to address the issue of the intrinsic hysteresis in the supplementary Figure S1.

4. Can the authors provide one set of data to show the magnetic field effect on T_c ?

The effect of magnetic field on T_c can be observed on Figure 3(a) and (b) – the superconducting transition shifts to lower temperature with increasing magnetic field. The high field data was obtained in 45 T hybrid magnet and in 65 T pulsed magnet by sweeping magnetic field at constant temperature. The limitations of these magnets precluded us from sweeping temperature at constant magnetic field. The traces of the temperature dependence of the sample resistance at fixed magnetic field values can be generated by taking constant-field slices of the raw data on Fig 3 (a) and (b). The field dependence of $T_c(H)$ will agree with the temperature dependence of $H_{c2}(T)$ on Fig 3(c).

5. Why the resistance of LaH10 under 138 GPa is much larger than the others measured between 120 GPa and 136 GPa?

The data of 138 GPa was obtained earlier than the other data, prior to sudden decrease of pressure down to 120 GPa during transportation of the sample to the High Field facility. We utilize van der Pauw geometry to measure the resistance. The electrical current path through the sample changed after the abrupt decompression (some amount of the surrounding H_2 medium and possibly LaH₁₀ sample could penetrate into the crack in diamond, which can result in the change of the LaH₁₀ geometry), thus the absolute value of the normal state resistance changed as well. However, the bulk superconducting parameters such as the T_c and H_c , are not affected.

6. What is the material for the dac? Did the authors use the same type of dac for the magnetotransport measurements and X-ray measurements?

The main material of the cell is CuTi alloy which has very low magnetic signal. We use the same miniature pressure cells for resistivity, magnetotransport and X-ray measurements. The prototype of this cell is published in Drozdov et al. (Nature 2015).

7. What is the rhenium gasket dimension? What is the sample space dimension?

The rhenium gasket was originally 250 μm thick and after the pre-indentation, insulation and application of pressure, the distance between the anvils is approximately 2-3 μm . The sample space is about 15 μm in diameter. The relevant text is modified in the method section.

8. The authors mentioned, “no apparent effect of the current value on the measured T_c was observed”. However, in the reference Phys. Rev. Lett. 122, 027001 (2019), M. Somayazulu et al. showed a clear effect of I on the T_c for LaH10. Could the authors comment on this difference?

We do not observe a significant dependence of T_c upon electrical current values, but we understand that detected critical current can vary significantly from sample to sample. The value

of the critical superconducting current in polycrystalline samples can be strongly affected by the weakest links in the electrical current path, particularly at the inter-grain junctions and at the contact with the electrical leads. Moreover, usually electrical leads have considerably high electrical resistance of at least about 100 Ω and can heat at high currents and change the real temperature of the superconducting sample in DAC. The bulk critical current density can be more accurately deduced from the magnetization measurements, however, these measurements are more challenging for microscopic samples in a massive metallic diamond anvil cell.

9. On the last paragraph of Page 4, the authors discussed phonon softening and coupling constant. Can the authors provide a more direct analysis on this part by using the detailed values related to LaH₁₀?

We calculated the phonon spectra in LaH₁₀. We did find a trend of the softening of certain phonons that makes the structure unstable on decompression. The results are detailed in the Supplementary Information.

REVIEWER COMMENTS

Reviewer #1 (Remarks to the Author):

The revised manuscript has improved the presentation of these experimental data. The authors also performed the suggested calculations with the support of their conclusions, The comparisons of the structural data between their synthesized samples and the early work from Geballe et al. have been done. The significance of this work has been addressed. The importance of the soft mode together with the structural instability for the enhancement of the superconducting transition temperature has been given. The revision of the manuscript is ready for the readers and can be recommended for the publication in Nature Communications. I would like to congratulate the authors for the contributions of this nice work on the superconducting superhydrides to the community.

Reviewer #2 (Remarks to the Author):

The questions raised previously were answered to a certain extent. However, some of the questions still need further explanation from the authors, as listed below:

1. The authors mentioned "The electrical transport properties and crystal structure of LaH10 at lower pressures were not carefully measured. Thus, the boundary and the role of structural instability were never known exactly. On the other hand, we found a 20 GPa difference in the pressure values between the previous study (Ref. 12) and the present data, because of the different pressure scales."
"1) What is the conclusion about the boundary and the role of structural instability based on this study?
2) What did the author mean by the difference is due to the "different pressure scales"?)
2. Can the authors elaborate on why the inclusion of quantum atomic fluctuations can stabilize the superconducting phase at lower pressure?
3. The authors pointed out that the pressure dependence of the T_c is independent of the pressurization history. For Fig. 2a, if I understand correctly, the first measurement was performed under 138 GPa, then the other measurements were done from 120 GPa to 136 GPa during the loading process (pressure increases), did the authors perform any measurements between 120 GPa and 138 GPa during unloading the cell?
4. What is the pressure for Fig. S1c? This curve is quite different from any of the curves shown in Fig. 2a. Can the authors explain? What causes the second transition (at lower T) in Fig. S1a and S1c?
5. The sample information was added this time. How about the size and arrangements of the leads? Fig. 2 showed that gold leads were used. Are they deposited on the diamond surface?
6. Regarding the authors' response to previous question #9, can the authors point out the specific part instead of just saying "the results are delated in the supplementary information"?

Reviewer #2 (Remarks to the Author):

The questions raised previously were answered to a certain extent. However, some of the questions still need further explanation from the authors, as listed below:

1. The authors mentioned “The electrical transport properties and crystal structure of LaH₁₀ at lower pressures were not carefully measured. Thus, the boundary and the role of structural instability were never known exactly. On the other hand, we found a 20 GPa difference in the pressure values between the previous study (Ref. 12) and the present data, because of the different pressure scales. “1) What is the conclusion about the boundary and the role of structural instability based on this study? 2) What did the author mean by the difference is due to the “different pressure scales”?)

1) The conclusion is stated in the last paragraph of the main text:

“We find evidence for a pressure-induced $Fm-3m - C2/m$ structural transition in LaH₁₀ at $p_c = 135$ GPa, resulting in a steep, but continuous decrease in $T_c(p)$ below p_c . A likely mechanism for the structural instability is phonon softening associated to a gradual distortion of the lattice, as proposed for another HTS hydride, namely H₃S.”

2) “different pressure scales” refers to the different pressure characterization techniques.

As detailed in the last reply to referee #1, the diamond scale often overestimates the pressure values by up to ~ 20 GPa in the studied pressure range.

2. Can the authors elaborate on why the inclusion of quantum atomic fluctuations can stabilize the superconducting phase at lower pressure?

As discussed in reference 14, the quantum fluctuations are essential to sustain crystals with large electron–phonon coupling constants, which is the case of LaH₁₀. The high symmetry $Fm-3m$ structure would otherwise be destabilized by a substantial electron–phonon interaction, resulting in distorted (low-symmetry) structures with lower electronic density of states at the Fermi level. The quantum effects are substantial, they reshape the energy landscape and stabilize structures by more than 60 meV per LaH₁₀.

We have added a brief elaboration on the matter as follows (highlighted in yellow):” ... the inclusion of the zero-point energy stemming from quantum atomic fluctuations lowers the enthalpy of the high-symmetry phase stabilizing it at pressures as low as 129 GPa.¹⁴”

3. The authors pointed out that the pressure dependence of the T_c is independent of the pressurization history. For Fig. 2a, if I understand correctly, the first measurement was performed under 138 GPa, then the other measurements were done from 120 GPa to 136 GPa during the loading process (pressure increases), did the authors perform any measurements between 120 GPa and 138 GPa during unloading the cell?

The sudden decompression from 138 GPa to 120 GPa was unexpected and occurred during the transportation of the cell to the High Magnetic Field facility. The answer to the above question is no. The temperature dependence of the resistance upon decompression was measured at end

values, i.e. at 138 and 120 GPa. Subsequent pressurization from 120 GPa to 136 GPa shows that similar pressures lead to similar T_c 's. In particular, the difference in T_c s is almost negligible before decompression and after the subsequent pressurizing: $T_c = 243$ K at 138 GPa and $T_c = 241$ K at 136 GPa. Thus, we believe that the pressure dependence of T_c is independent on the pressurization history.

4. What is the pressure for Fig. S1c? This curve is quite different from any of the curves shown in Fig. 2a. Can the authors explain? What causes the second transition (at lower T) in Fig. S1a and S1c?

The curves in Fig. S1c are obtained in a dedicated study of the raw data from a prior publication [Ref S1]. The study was prompted by the apparent hysteresis present in the earlier HTS hydride reports and the claims of its intrinsic nature. The curves represent the superconducting transition in H_3S HTS hydride sample and illustrate the hysteresis in temperature between cooling and warming cycles, as registered through sensors attached to the different parts of the DAC. The curves shown in the main text are measured with LaH_{10} . The names and conditions of the sample have been added to the plots in Fig. S1. The second transition (at lower T_s) seems to come from sample or pressure inhomogeneity, for example, where a portion of the sample stays at lower pressures [Ref S1]

We edited the caption for figure S1 for clarity.

5. The sample information was added this time. How about the size and arrangements of the leads? Fig. 2 showed that gold leads were used. Are they deposited on the diamond surface?

The following was added to the methods section as highlighted in blue:

"The 120-nm-thick tantalum leads covered by a 50-nm-thick gold layer were sputtered onto the diamond surface in a van der Pauw configuration with a distance of $\sim 4 \mu m$ between tips (see Supplementary Fig. S7 inserts)."

6. Regarding the authors' response to previous question #9, can the authors point out the specific part instead of just saying "the results are delated in the supplementary information"?

We added the following to the main text, highlighted in green:

"The calculated phonon dispersions show that the $Fm-3m$ phase is dynamically stable at 200 GPa. However, softening of the low-lying H-H "wagging" vibration modes along the G-X direction are found in the phonon spectrum (Supplementary Fig. S4, S5), which leads to a structural instability towards the monoclinic $C2/m$ distortion. The classical harmonic treatment of atomic vibrations for the $Fm-3m$ phase of LaH_{10} shows negative phonon frequencies at pressures below 180 GPa. Besides, a boost in T_c due to phonon softening in the vicinity of a structural transition has been widely reported."

REVIEWERS' COMMENTS

Reviewer #2 (Remarks to the Author):

My questions were addressed properly and I recommend the publication of this work in Nature Communications.